# Perceived Organizational Culture and Turnover Intentions: The Serial Mediating Effect of Perceived Organizational Support and Job Insecurity

**Mónica Salvador [1], Ana Moreira [2,*] and Liliana Pitacho [3,4]**

1 Department of Psychology, Instituto Superior Manuel Teixeira Gomes, 8500-590 Portimão, Portugal
2 School of Psychology, ISPA—Instituto Universitário, Rua do Jardim do Tabaco 34, 1149-041 Lisboa, Portugal
3 Escola Superior Ciências Empresariais, Instituto Politécnico de Setúbal, Campus do IPS—Estefanilha, 2910-761 Setúbal, Portugal
4 Centro de Administração e Políticas Públicas, Instituto Superior de Ciências Sociais e Políticas, Universidade de Lisboa, 1300-663 Lisboa, Portugal
* Correspondence: anasetemoreira@gmail.com or amoreira@ispa.pt

**Abstract:** This study aims to analyze the relationship between perceived organizational culture (POC) and turnover intentions (TI) and if this relationship is mediated by perceived organizational support (POS) and job insecurity (JI). For this purpose, the following hypotheses were formulated: (1) POC (support, goals, innovation, and rules) has a negative and significant relationship with TI; (2) POC (support, goals, innovation, and rules) has a positive and significant relationship with POS (affective and cognitive); (3) POS (affective and cognitive) has a negative and significant relationship with TI; (4) POS (affective and cognitive) has a negative and significant relationship with JI; (5) JI has a positive and significant relationship with TI; and (6) POS (affective and cognitive) and JI both represent a serial indirect effect in the relationship between POC (support, goals, innovation and rules) and the TI. This study's sample includes 661 participants working in organizations based in Portugal. The results indicate that only the perception of supportive and goal culture has a negative and significant association with TI; POC has a positive and significant association with POS; POS has a negative and significant association with JI and TI; JI has a positive and significant association with TI; affective POS and JI have a serial mediation effect in the relationship between supportive and goal POC and TI; cognitive POS and JI have a serial mediation effect in the relationship between goal POC and TI.

**Keywords:** perceived organizational culture; turnover intentions; perceived organizational support; job insecurity

## 1. Introduction

Although telework began to be used in the 1970s, it was not until the COVID-19 pandemic that its use expanded worldwide.

Today, telecommuting is under increasing discussion due to its widespread use worldwide because of COVID-19. Due to the advancement of technology, working outside of offices has become increasingly common and attractive to many people (Allen et al. 2015). The transition from a physical work environment to a virtual work environment can bring benefits and limitations for the organization as a whole. One of these limitations refers to the innovation culture (Allen et al. 2015). This is one of the reasons that led us to study the association between POC and other organizational variables at a time when many work teams are still in a hybrid model. In this study, we will address two theories of culture: Schein's theory and Cameron and Quinn's theory. Schein's theory will be addressed due to its relevance in interpreting the relationship between culture and the perception of organizational support. Cameron and Quinn's theory will be addressed and explained because the questionnaire applied to assess culture is based on this theory.

TI is one of the most fragile points in the organizational context since leaving the organization has a high cost for it, making it extremely relevant to study the relationship with its antecedents (Reiche 2008). Among the antecedents of TI, we have POC and POS: that is, when employees have low perceptions of these variables, their perception of JI becomes high, and TI increases (Brandão et al. 2012; Kim and Jang 2018).

As for JI, it is related to several negative consequences and implications for both the employee and the organization, one of these consequences being TI (Richter et al. 2020). In turn, POC is considered by Kim and Jang (2018) as one of the antecedents of POS.

The main objective of this study is to study whether the perception of a better organizational culture reduces the intentions to leave the organization. At the same time, and due to gaps in the research relating to the four variables under study, we tried to understand whether the POS and JI are mechanisms that explain the relationship between POC and TI. As specific objectives, we intend to study the relationship between the various variables under study.

## 1.1. Perceived Organizational Culture

Organizational culture is defined by the set of attitudes, norms, values, assumptions and beliefs that are shared by members of the organization (Stone et al. 2007). This can be operationalized through the collective behaviors that develop based on the interactions of individuals in an organization (Kusmaul and Sahoo 2019).

According to Schein (1999), there are several levels of culture, some being more conscious than others. Culture is reflected through formal policies, strategic intentions and the internal structure of the organization. Thus, it is important to understand the perception of employees regarding organizational culture and to understand its implication in other variables of organizational relevance. Schein (1999) defines the organization as a structure or social reality and culture as a historical phenomenon that is changeable and is an integrating and stabilizing force in the organizational context. In his theory, Schein (1999) postulates three levels in organizational culture: the most superficial level is where the artefacts are found, which are the shared norms, a more concrete and more observable component, where these norms function as behavioral guidelines; the second level, which is an intermediate level, is where the values are found, which are common and which shape the essence of the organization conferring the sense of identity to it because when they are identified by the employees, they feel part of the organization; and finally, the last level of the model is the base assumptions, which are considered to be acquired, invisible and present in an unconscious way.

Another model to take into consideration when we address the organizational culture is the model of contracting values (Quinn and Cameron 1983). In this model, the variables taken into account are control and flexibility and internal and external focus, and four types of culture are identified: the supportive or clan culture, the innovative or adhocracy culture, the bureaucratic or hierarchy culture, and the rational or market culture. The supportive culture is characterized by shared values and goals, high teamwork, employee involvement in organizational programs and shared commitments, employee empowerment and facilitated employee participation. The innovation culture is characterized by being temporary and dynamic, requires a high level of expertise from employees, and employees are willing to deal with change and new challenges based on organizational flexibility and creativity. The bureaucratic culture is characterized by a well-defined hierarchy, separation of powers, and impersonality, there is a standardization of processes, and it is based on formal rules and policies, with a division of labor and only formal relationships among employees. Finally, the rational culture is characterized by an external focus; the goal is to create competitive advantages and achieve goals. It is a results-oriented culture and competition based on the achievement of goals; this culture is directed and oriented to customers and their needs.

The organizational culture is based on the behaviors and actions of employees who are part of the organization, reflecting how they behave on a daily basis. Culture is verified

in the context of professional activity, and it is in this context that the adaptive skills that will influence compliance with collective rules are developed (Ventura et al. 2020).

*1.2. Turnover Intentions*

Turnover intentions correspond to the willingness that employees have to leave the organization where they are and start the search for a new workplace (Tett and Meyer 1993; Benson 2006). For Basariya and Ahmed (2019), TI occurs when the employee plans to leave their job, but their turnover has not yet been effected; it is only pondered and planned. However, Tett and Meyer (1993) consider that TI does not always result in an effective exit from the organization.

The antecedents of TI may have individual or organizational characteristics, such as the perception of organizational support (Hui et al. 2007), the organizational culture (Islam et al. 2012), and the perception of human resource management practices (Heavey et al. 2013). According to Shore and Martin (1989), the antecedents of TIs can also be attitudinal, such as organizational commitment and job satisfaction. In this study, we studied some of the antecedents of TI, such as POC, POS, and JI.

In a study in Taiwan, Wang et al. (2012) found that turnover intentions differed between public and private sector employees, with private sector employees showing higher turnover intentions than public sector employees. The same results were obtained in a study conducted in Serbia by Mihajlov and Mihajlov (2016).

According to Hom et al. (2017), several strategies can enhance employee retention and decrease TI, such as effective recruitment, effective leadership by managers, development of employee skills, employee satisfaction, organizational culture, and balance between family and work. These aspects have great emphasis and relevance to the construct described above.

1.2.1. Perceived Organizational Culture and Turnover Intentions

The interpretation and perception of organizational culture are identified as factors related to organizational turnover intentions (Ivanova 2019). Additionally, according to Islam et al. (2012), POC has a negative and significant association with TI, thus acting as a reducer.

Pinto (2019) found that if POC is negative, it can lead to a higher absenteeism rate and increased TI. According to Ivanova (2019), several factors influence TI, including mismatch at the organizational culture level. This relationship can be interpreted based on the theory of resource conservation (Hobfoll 1989), according to which employees seek to create, protect, and maintain labor resources whose possession will lead them to positive outcomes, and, naturally, an employee with a high POC does not want to lose that resource, decreasing their TI. This is the reasoning that leads us to formulate the following hypothesis.

**Hypothesis 1 (H1).** *POC (support, goals, innovation, and rules) has a negative and significant relationship with TI.*

*1.3. Perceived Organizational Support*

Perceived organizational support is defined as the perception that the organization's human resources have of the employees' values and beliefs and how the organization cares about their well-being (Eisenberger et al. 1986). This factor contributes to employees' well-being, motivation, and quality of life (Tomasi et al. 2020).

Justice, management support, and human resource practices are antecedents that, when correctly applied, increase the perception of organizational support. Perceived organizational support also has some relevant aspects as consequences, such as affective organizational commitment, job satisfaction, organizational citizenship behaviors,

performance, intentions to stay in and leave the organization, organizational trust, and absenteeism (Rhoades and Eisenberger 2002; Muneer et al. 2014; Kurtessis et al. 2015).

### 1.3.1. Perceived Organizational Culture and Perceived Organizational Support

According to Van Beek and Gerritsen (2010), personal perceptions of organizational culture can be seen as an indicator of the quality of organizational support. Lee et al. (2013) claim that perceptions of organizational culture are strongly related to employees' commitment to the organization. A study conducted with ship employees found POC to positively affect POS and organizational support (Kim and Jang 2018). According to Berson et al. (2009), the perception of organizational culture is an antecedent of the perception of organizational support. This relationship can be interpreted in light of the social identity theory (Tajfel 1978), according to which employees perceive themselves as members of the organization to which they belong, absorbing its characteristics such as POS and reducing their TI. The hypothesis is formulated as follows:

**Hypothesis 2 (H2).** *POC (support, goals, innovation, and rules) has a positive and significant relationship with POS (affective and cognitive).*

### 1.3.2. Perceived Organizational Support and Turnover Intentions

From the perspective of Hui et al. (2007), POS is one of the antecedents of TI. In a study conducted by Wang et al. (2020), in a sample of employees who are in contact with the customer, the results also indicate that when employees perceive high support from their organization, they try to reciprocate this perception by staying in the organization. Additionally, Jing and Yan (2022), in a study conducted in China, concluded that POS has a significant and negative effect on TI. This relationship can be explained by the norm of reciprocity (Gouldner 1960) and the premise of social exchanges (Blau 1964). This relationship can also be explained by the social comparison theory (Adams 1965). Employees tend to compare their organizations with others, and if they perceive that their organizations offer better organizational support than others, their TI decrease. The following hypothesis was thus formulated:

**Hypothesis 3 (H3).** *POS (affective and cognitive) has a negative and significant relationship with TI.*

### 1.4. Job Insecurity

Job insecurity is based on the perceptions and interpretations that the subject has about the work environment in which he/she is inserted (Sverke et al. 2006). According to Grunberg et al. (2006), JI can be defined as the worker's perception of the threat of involuntary loss of his or her job in the near future.

According to De Witte (2005), JI can arise from several antecedents that are related to three levels: the macro level that relates to the region and/or the organization; the individual level that is characterized by the worker's position in the organization; personality traits that relate to the locus of control and negative affectivity. Within these levels, there are more variables that relate to the antecedents of JI. Job insecurity has several consequences that include lower well-being and health (De Witte et al. 2015), breach of the psychological contract (Ma et al. 2019), lower affective commitment (Çakmak-Otluoğlu and Ünsal-Akbiyik 2015), decreased performance (Piccoli et al. 2019), decreased organizational citizenship behaviors, breach of trust, increased resistance to change, and increased TI (Obeng et al. 2020).

### 1.4.1. Perceived Organizational Support and Job Insecurity

According to Bohle et al. (2018), POS is a factor that plays a significant role in employees' JI. POS is linked to job stability, so if there is a low level of POS, there is a high

level of JI, and there is an inverse proportionality relationship (Brandão et al. 2012). When an organization is concerned with the support it provides to its employees, such as the development of their skills, according to the human capital theory (Schultz 1961), at the same time that it invests in these skills, it transmits to them a lower perception of JI. Thus, the following hypothesis was formulated.

**Hypothesis 4 (H4).** *POS (affective and cognitive) has a negative and significant relationship with JI.*

1.4.2. Job Insecurity and Turnover Intentions

Greenhalgh and Rosenblatt (1984) state that employees with high levels of JI more often seek new jobs to increase job security; this is more intensely seen in employees with higher qualifications. According to Mobley (2011), when high levels of JI are present, TI tends to increase as a consequence. In a study by Obeng et al. (2020), the results also show that JI has a positive and significant relationship with TI. When they perceive high JI in their organization, employees tend to have high TI, a relationship explained by the premise of social exchanges (Blau 1964) because if their organization does not allow them to have job security, in exchange, their TI increases. The following hypothesis was thus formulated.

**Hypothesis 5 (H5).** *JI has a positive and significant relationship with TI.*

*1.5. Perceived Organizational Culture, Perceived Organizational Support, Job Insecurity and Turnover Intentions*

Although there have been several studies on the above variables and their relationships, no studies that relate the four variables simultaneously have been found, so the research is relevant.

Based on the relationships studied above, we expect that POC will positively influence POS (Kim and Jang 2018), and POS will negatively influence JI (Bohle et al. 2018), which in turn will potentiate TI (Obeng et al. 2020). This reasoning leads us to propose the following hypothesis.

**Hypothesis 6 (H6).** *POS (affective and cognitive) and JI, both represent a serial indirect effect in the relationship between POC (support, goals, innovation and rules) and the TI.*

**Hypothesis 6a (H6a).** *POS (affective and cognitive) and JI both represent a serial indirect effect in the relationship between support POC and the TI.*

**Hypothesis 6b (H6b).** *POS (affective and cognitive) and JI both represent a serial indirect effect in the relationship between goal POC and the TI.*

**Hypothesis 6c (H6c).** *POS (affective and cognitive) and JI both represent a serial indirect effect in the relationship between innovation POC and the TI.*

**Hypothesis 6d (H6d).** *POS (affective and cognitive) and JI both represent a serial indirect effect in the relationship between rules POC and the TI.*

In order to integrate the hypotheses formulated in this study, the following theoretical model was developed, where the expected relations are synthesized (Figure 1).

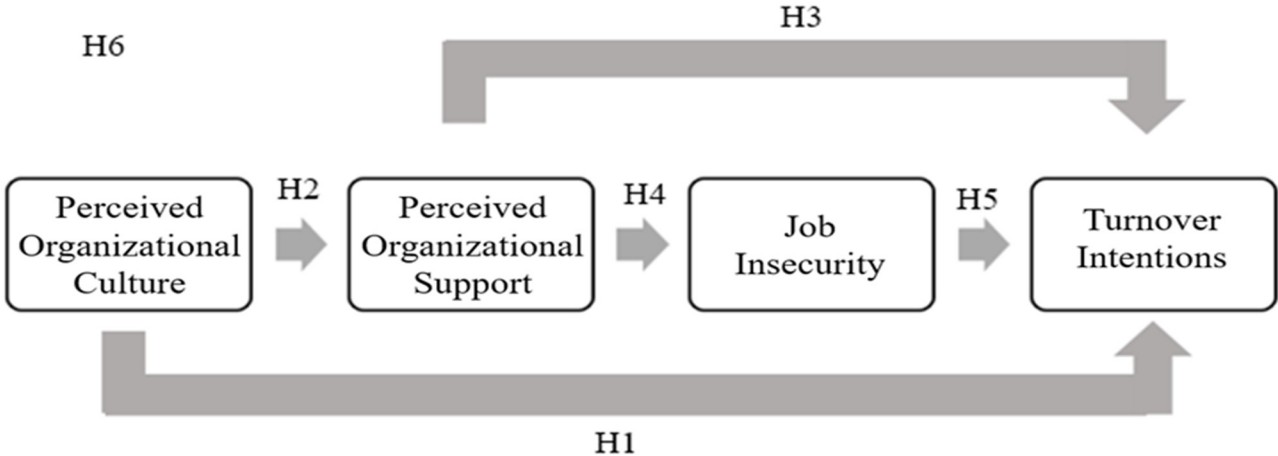

**Figure 1.** Research model.

## 2. Methods

### 2.1. Procedure

A total of 661 participants collaborated voluntarily in this study, all of which were considered in the following statistical analyses because they met the conditions for participation in this study (they were currently working in organizations in Portugal). The sampling process was non-probability, convenience and intentional snowball type (Trochim 2000).

The questionnaire that was placed online on the Google Forms platform contained information about the purpose of the study. The questionnaire also expressed that the confidentiality of responses would be maintained. The questionnaire included seven questions for sample characterization (age, gender, academic qualifications, region of Portugal where they live, seniority in the organization, seniority in the position and work sector) and four scales (perceived organizational culture, perceived organizational support, job insecurity and turnover intentions). Data were collected between February 2022 and April 2022.

### 2.2. Participants

A descriptive analysis of the sample shows a sample of 661 participants aged between 19 and 67 years (M = 42.51; SD = 11.42), with 449 female participants (67.9%) and 212 male participants (32.1%). About the participants' academic qualifications, 191 (28.9%) have 12th grade or less, 333 (50.4%) have an undergraduate degree, and 137 (20.7%) have a master's degree or higher. Regarding the region of Portugal where they live, 198 (30%) live in the north region, 138 (20.9%) live in the Lisbon metropolitan area, 126 (19.1%) live in the Algarve region, 116 (17.5%) live in the central region, 54 (8.2%) live in the Alentejo, 19 (2.9%) live in the Azores, and 10 (1.5%) live in Madeira. The seniority in the organization varies between 0.25 and 43 years (M = 12.05; SD = 11.21) and seniority in the job varies between 0.25 and 44 years (M = 11.61; SD = 10.50). Regarding the work sector, 412 (62.3%) participants work in the public sector and 249 (37.7%) in the private sector.

### 2.3. Data Analysis Procedure

The data were imported into a database in SPSS Statistics 28 software (IBM Corp., Armonk, NY, USA). The first analysis was related to the instruments, testing their metric qualities. The validity of the instruments was tested by performing confirmatory factor analyses using AMOS 28 for Windows software (IBM Corp., Armonk, NY, USA). The procedure was according to a "model generation" logic (Jöreskog and Sörbom 1993), considering in the analysis of their adjustment the results obtained for the chi-square ($\chi^2$/df) $\leq$ 5; for the Tucker–Lewis index (TLI) > 0.90; for the goodness-of-fit index (GFI) > 0.90; for

the comparative fit index (CFI) > 0.90; for the root mean square error of approximation (RMSEA) ≤ 0.08; and for the root mean square residual (RMSR), a smaller value corresponds to a better adjustment.

Next, the internal consistency of the scale was analyzed by calculating Cronbach's alpha, whose value should vary between "0" and "1", not assuming negative values (Hill and Hill 2002) and being higher than 0.70, the minimum acceptable in organizational studies (Bryman and Cramer 2003). Composite reliability was also calculated for each dimension of the instruments. The average variance extracted (AVE) was calculated to test the convergent validity. Concerning sensitivity, measures of central tendency, dispersion and distribution were calculated for the items of the scales used, thus performing the normality study for all items and all scales.

Hypotheses 1, 2, 3, 4, and 5 formulated in this study were tested using simple and multiple linear regressions. To test the mediation model, hypothesis 6, we used the PRO-CESS 4.0 macro (Hayes, New York, NY, USA), developed by Hayes (2013), since it allows us to test a mediation model with multiple mediators operating in series.

*2.4. Instruments*

The instruments used in this study are presented below. These instruments were chosen since they are all validated for the Portuguese population.

To assess the POC variable, we used only the organizational culture subscale of the FOCUS (first organizational culture unified search) instrument, validated and adapted for the Portuguese population by Neves (2000), composed of 35 items with a 6-point Likert type response (from 1 "Not at all" to 6 "Very much"). This subscale is composed of four dimensions corresponding to the four types of culture of the Contrasting Values Model: innovation culture (items 1, 10, 14, 17, 30, 32, and 34); support culture (items 2, 7, 16, 19, 21, 25, 26, 28, 29, 33); goal culture (items 3, 6, 8, 11, 13, 22, 23, 31); rule culture (items 4, 5, 9, 12, 15, 18, 20, 24, 27, 35). The validity was tested through a four-factor confirmatory factor analysis, which confirmed their existence. It should be noted that items 1, 27 and 30 had to be removed because they had a low factor weight. The adjustment indexes obtained were adequate ($\chi^2$/df = 3.90; GFI = 0.86; CFI = 0.92; TLI = 0.91; RMSEA = 0.066; SRMR = 0.089). As for internal consistency, Cronbach's alpha was 0.79 for innovation culture, 0.95 for support culture, 0.90 for goal culture, and 0.90 for rule culture. About the composite reliability, the values vary between 0.77 (innovation culture) and 0.95 (support culture). For convergent validity, values between 0.42 (innovation culture) and 0.64 (supportive culture) were obtained.

Regarding the TI construct, the scale used was developed by Bozeman and Perrewé (2001) and translated and adapted to the Portuguese population by Bártolo-Ribeiro (2018). The scale is composed of 6 items. The response is given through a 5-point Likert-type scale (from 1 "Does not apply at all" to 5 "Totally applies"). Its validity was tested through a one-factor confirmatory factor analysis, which confirmed its existence. The adjustment indexes obtained are adequate ($\chi^2$/gl = 1.79; GFI = 0.99; CFI = 0.99; TLI = 0.99; RMSEA = 0.035; SRMR = 0.018). As for internal consistency, it presents a Cronbach's alpha of 0.93. This instrument has a composite reliability of 0.92 and convergent validity of 0.66.

The SPOS (Survey Perceived Organization Support) scale developed by Eisenberger et al. (1997) was used to infer results concerning POS. It was adapted and validated for the Portuguese population by Santos and Gonçalves (2010). The scale is composed of 8 items with a 7-point Likert-type response (from 1 "Strongly Disagree" to 7 "Strongly Agree"). This instrument is composed of two dimensions: cognitive POS (items 1, 4, 6 and 8) and cognitive POS (items 2, 3, 5 and 7). Its validity was tested through a two-factor confirmatory factor analysis, which confirmed its existence. The adjustment indexes obtained are adequate ($\chi^2$/gl = 2.43; GFI = 0.99; CFI = 0.99; TLI = 0.99; RMSEA = 0.047; SMRM = 0.089). As for internal consistency, a Cronbach's alpha of 0.93 was obtained for the affective POS and 0.87 for the cognitive POS. The composite reliability has a value of

0.93 for the affective POS and 0.87 for the cognitive POS. In turn, convergent validity has a value of 0.77 for the cognitive POS and 0.63 for the cognitive POS.

The Job Insecurity Scale (De Witte 2000) was applied to assess JI, consisting of 8 items. The items are rated on a 5-point Likert-type scale (From 1 "Strongly Disagree" to 5 "Strongly Agree"). This instrument is composed of two dimensions: qualitative JI (items 2, 3, 6 and 8) and quantitative JI (items 1, 4, 5 and 7). Its validity was tested using a two-factor confirmatory factor analysis, which was not found to be the case. The two factors were strongly correlated and not all adjustment indices proved to be adequate ($\chi^2$/gl = 13.90; GFI = 0.93; CFI = 0.95; TLI = 0.91; RMSEA = 0.140; SRMR = 0.142). Subsequently, one-factor confirmatory factor analysis was performed, which confirmed the existence of only one factor. The adjustment indexes obtained were adequate ($\chi^2$/gl = 3.22; GFI = 0.99; CFI = 0.99; TLI = 0.99; RMSEA = 0.058; SRMR = 0.026). As for internal consistency, a Cronbach's alpha of 0.90 was obtained. Composite reliability has a value of 0.86 and convergent validity of 0.47.

Neither the instruments nor their component items grossly violate normality.

## 3. Results

### 3.1. Descriptive Statistics of the Variables under Study

The first step was to perform descriptive statistics of the variables under study in order to understand the answers given by the participants. Participants of this study were found to have a perceived culture of innovation (t (660) = 7.05; $p < 0.001$), support (t (660) = 16.08; $p < 0.001$), goals (t (660) = 17.29; $p < 0.001$), and rules (t (660) = 21.68; $p < 0.001$), significantly above the central point (3.5) (Table 1 and Figure 2). It should be noted that the least perceived culture is the culture of innovation. We also tried to understand if the perception of culture varies depending on the area where the employee works. It was found that in Alentejo and Azores, the type of culture with the highest perception is the supportive culture. In all other regions, the type of culture with the highest perception is the rules culture.

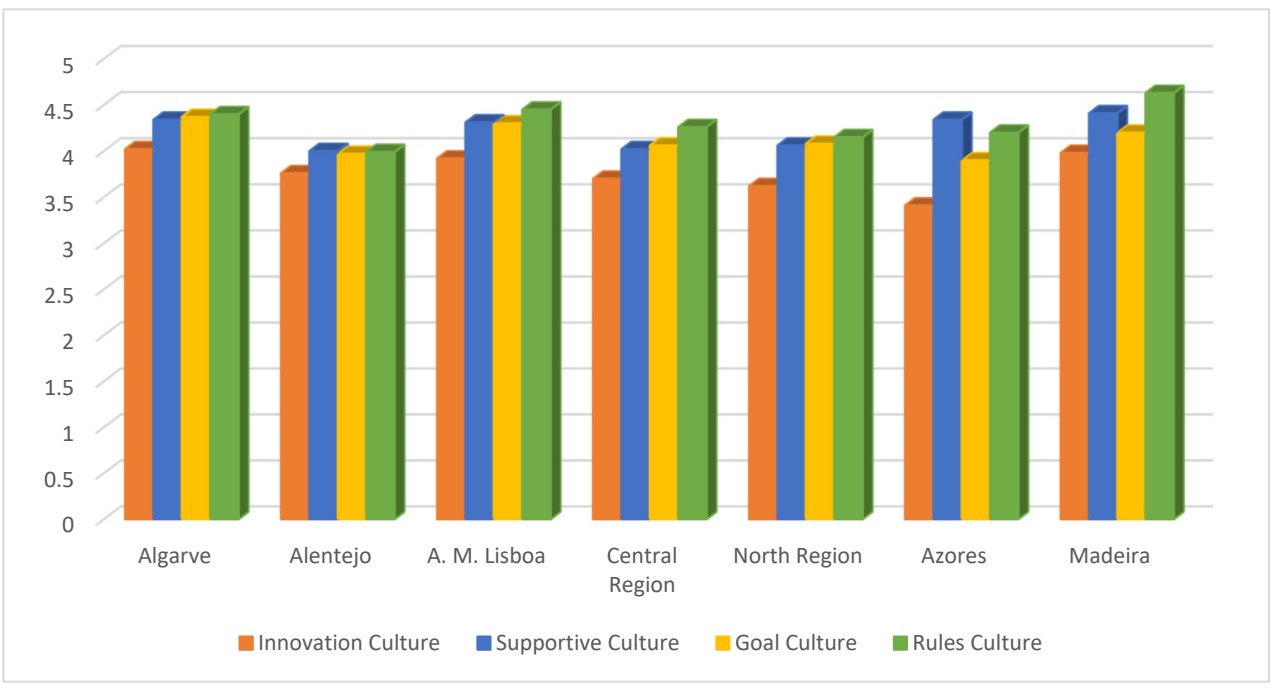

**Figure 2.** Distribution of culture type according to the region of Portugal.

**Table 1.** Means, standard deviation and association between the variables under study.

|  | Mean | SD | 1.1 | 1.2 | 1.3 | 1.4 | 2.1 | 2.2 | 3 | 4 |
|---|---|---|---|---|---|---|---|---|---|---|
| 1.1. Innovation Culture | 3.78 | 1.04 | – |  |  |  |  |  |  |  |
| 1.2. Supportive Culture | 4.17 | 1.07 | 0.74 *** | – |  |  |  |  |  |  |
| 1.3. Goal Culture | 4.17 | 1.00 | 0.79 *** | 0.80 *** | – |  |  |  |  |  |
| 1.4. Rules Culture | 4.27 | 0.92 | 0.56 *** | 0.59 *** | 0.76 *** | – |  |  |  |  |
| 2.1. Affective POS | 4.32 | 1.60 | 0.68 *** | 0.74 *** | 0.70 *** | 0.49 *** | – |  |  |  |
| 2.2. Cognitive POS | 4.42 | 1.62 | 0.38 *** | 0.45 *** | 0.41 *** | 0.24 *** | 0.60 *** | – |  |  |
| 3. Job Insecurity | 2.16 | 0.93 | −0.24 *** | −0.33 *** | −0.27 *** | −0.17 *** | −0.37 *** | −0.39 *** | – |  |
| 4. Turnover Intentions | 2.23 | 1.16 | −0.28 *** | −0.34 *** | −0.34 *** | −0.26 *** | −0.46 *** | −0.38 *** | 0.45 *** | – |

Note. *** $p < 0.001$.

The participants in this study were also found to have high affective (t (660) = 5.14; $p < 0.001$) and cognitive (t (660) = 6.70; $p < 0.001$) POS, above the central point of the scale (4) (Table 1). As for both JI (t (660) = −17.06; $p < 0.001$) and TI (t (660) = −23.34 $p < 0.001$), on the other hand, the responses given by the participants lie below the central point of the scales (3) (Table 1).

Next, to understand whether the activity sector (public and private) had a significant effect on the variables under study, several *t*-student tests for independent samples were performed.

The activity sector was found to have a significant effect on POC (innovation, support, goals and rules), as well as affective POS and TI. Participants in the private sector have a higher perception of POC and affective POS and higher TI (Table 2).

**Table 2.** Results of the *t*-student tests for independent samples.

| Dependent Variable | *t* | *p* | Private | | Public | |
|---|---|---|---|---|---|---|
|  |  |  | Mean | SD | Mean | SD |
| Innovation Culture | 5.96 *** | <0.001 | 4.09 | 1.09 | 3.60 | 0.96 |
| Support Culture | 3.75 *** | <0.001 | 4.37 | 1.04 | 4.05 | 1.07 |
| Rules Culture | 2.32 * | 0.021 | 4.38 | 0.85 | 4.21 | 0.95 |
| Goals Culture | 3.83 *** | <0.001 | 4.34 | 0.89 | 4.06 | 1.03 |
| Affective POS | 2.85 ** | 0.005 | 4.55 | 1.70 | 4.18 | 1.52 |
| Cognitive POS | 1.81 | 0.071 | 4.57 | 1.76 | 4.33 | 1.52 |
| Job Insecurity | 1.20 | 0.232 | 2.22 | 0.98 | 2.13 | 0.89 |
| Turnover Intentions | 5.84 *** | <0.001 | 2.57 | 1.21 | 2.02 | 1.07 |

Note. * $p < 0.05$; ** $p < 0.01$; *** $p < 0.001$.

### 3.2. Association between the Variables under Study

Next, Pearson's correlations were performed to study the variables' association. TI and JI are negatively and significantly correlated with all culture types and both affective and cognitive POS. The association between TI and JI was positive and significant. Affective and cognitive POS are positively and significantly associated with all organizational culture types (Table 1).

### 3.3. Hypothesis Tests

To test Hypothesis 1, a multiple linear regression was performed. Results are shown in Table 3.

**Table 3.** Multiple linear regression results (H1).

| Independent Variables | Dependent Variable | F | *p* | $R^2_a$ | β | *p* |
|---|---|---|---|---|---|---|
| Innovation Culture | | | | | 0.02 | 0.788 |
| Supportive Culture | Turnover Intentions | 23.74 | <0.001 | 0.12 | −0.19 ** | 0.002 |
| Goal Culture | | | | | −0.18 * | 0.034 |
| Rules Culture | | | | | −0.03 | 0.427 |

Note. * *p* < 0.05; ** *p* < 0.01.

The results indicate to us that only supportive culture (β = −0.19; *p* = 0.002) and goals culture (β = −0.18; *p* = 0.034) have a negative and significant effect on TI. The model explains 12% of the variability in the dependent variable and is statistically significant (F (4, 656) = 23.74; *p* < 0.001).

Two multiple linear regressions were performed to test Hypothesis 2. The results are presented in Table 4.

**Table 4.** Results of the multiple linear regressions (H2).

| Independent Variables | Dependent Variable | F | *p* | $R^2_a$ | β | *p* |
|---|---|---|---|---|---|---|
| Innovation Culture | | | | | 0.19 *** | <0.001 |
| Supportive Culture | Affective POS | 245.96 *** | <0.001 | 0.60 | 0.43 *** | <0.001 |
| Goal Culture | | | | | 0.27 *** | <0.001 |
| Rules Culture | | | | | −0.08 * | 0.042 |
| Innovation Culture | | | | | 0.05 | 0.369 |
| Supportive Culture | Cognitive POS | 45.87 *** | <0.001 | 0.21 | 0.31 *** | <0.001 |
| Goal Culture | | | | | 0.23 ** | 0.004 |
| Rules Culture | | | | | −0.14 ** | 0.008 |

Note. * *p* < 0.05; ** *p* < 0.01; *** *p* < 0.001.

Innovation culture (β = 0.19; *p* < 0.001), support culture (β = 0.43; *p* < 0.001), and goals culture (β = 0.27; *p* < 0.001) have a positive and significant effect on affective POS. Rule culture (β = −0.08; *p* = 0.042) has a negative and significant effect on affective POS. The model explains 60% of the variability in the dependent variable and is statistically significant (F (4, 656) = 245.96; *p* < 0.001).

The results further indicate to us that only supportive culture (β = 0.31; *p* < 0.001) and goal culture (β = 0.23; *p* = 0.004) have a positive and significant effect on cognitive POS. Rule culture (β = −0.14; *p* = 0.004) has a negative and significant effect on cognitive POS. The model explains 21% of the variability in the dependent variable and is statistically significant (F (4, 656) = 45.87; *p* < 0.001).

Multiple linear regression was performed to test Hypothesis 3. The results are presented in Table 5.

**Table 5.** Multiple linear regression results (H3).

| Independent variables | Dependent Variable | F | *p* | $R^2_a$ | β | *p* |
|---|---|---|---|---|---|---|
| Affective POS | Turnover Intentions | 94.97 *** | <0.001 | 0.22 | −0.36 *** | <0.001 |
| Cognitive POS | | | | | −0.16 *** | <0.001 |

Note. *** *p* < 0.001.

The results indicate to us that both affective (β = −0.36; *p* < 0.001) and cognitive (β = −0.16; *p* < 0.001) POS have a negative and significant effect on TI. The model explains 22% of the variability in the dependent variable and is statistically significant (F (2, 658) = 94.97; *p* < 0.001).

In order to test Hypothesis 4, a multiple linear regression was performed, and the results are shown in Table 6.

**Table 6.** Multiple linear regression results (H4).

| Independent variables | Dependent Variable | F | $p$ | $R^2_a$ | β | $p$ |
|---|---|---|---|---|---|---|
| Affective POS | Job Insecurity | 71.44 *** | <0.001 | 0.18 | −0.21 *** | <0.001 |
| Cognitive POS | | | | | −0.27 *** | <0.001 |

Note. *** $p < 0.001$.

Both affective (β = −0.21; $p < 0.001$) and cognitive (β = −0.27; $p < 0.001$) PSO have a negative and significant effect on JI. The model explains 18% of the variability in the dependent variable and is statistically significant (F (2, 658) = 71.44; $p < 0.001$).

Hypothesis 5 was tested using simple linear regression, of which the results are shown in Table 7.

**Table 7.** Simple linear regression results (H5).

| Independent variable | Dependent Variable | F | $p$ | $R^2$ | β | $p$ |
|---|---|---|---|---|---|---|
| Job Insecurity | Turnover Intentions | 166.33 *** | <0.001 | 0.20 *** | 0.45 | <0.001 |

Note. *** $p < 0.001$.

The results indicate that JI (β = 0.45; $p = 0.002$) has a positive and significant effect on TI. The model explains 20% of the variability in the dependent variable and is statistically significant (F (1, 659) = 166.33; $p < 0.001$).

Hypothesis 6 states that POS and JI represent a serial indirect effect in the relationship between POC and TI. The procedures recommended by Baron and Kenny (1986) were followed to test this hypothesis. For this reason, only hypotheses 6a and 6b were tested. Specifically, Models 1 and 2 are the results of Hypothesis 6a tests, models 3 and 4 of Hypothesis 6b.

When testing Hypothesis 6a, a significant total indirect effect was observed since the confidence interval did not contain zero. This indirect effect is divided into three indirect effects, again all significant: the serial indirect effect; the indirect effect in which affective POS mediates the relationship between supportive culture and TI; the indirect effect in which JI mediates the relationship between supportive culture and TI (Table 8). When the contrasts were analyzed, it was found that the strongest indirect effect is the one in which affective POS mediates the relationship between supportive culture and TI. When the mediators were introduced in the regression equation, the direct effect of supportive culture on TI ceased to be significant, which leads to the conclusion that we are dealing with a total mediation effect (Figure 3).

**Table 8.** Indirect effects of Model 1.

| | Indirect Effects | |
|---|---|---|
| | Estimates | 95% CI with Bootstrap Correction |
| Model 1 | | |
| Total | −0.41 (0.05) | [−0.50; −0.32] |
| SC → APOS → TI | −0.29 (0.04) | [−0.38; −0.21] |
| SC → JI → TI | −0.04 (0.02) | [−0.09; −0.01] |
| SC → APOS → JI → TI | −0.07 (0.02) | [−0.11; −0.04] |

Note. Total Effect SC → TI = −0.36 (0.04). The standard error is in parentheses. SC = supportive culture; TI = turnover intentions; APOS = affective perceived organizational support; JI = job insecurity.

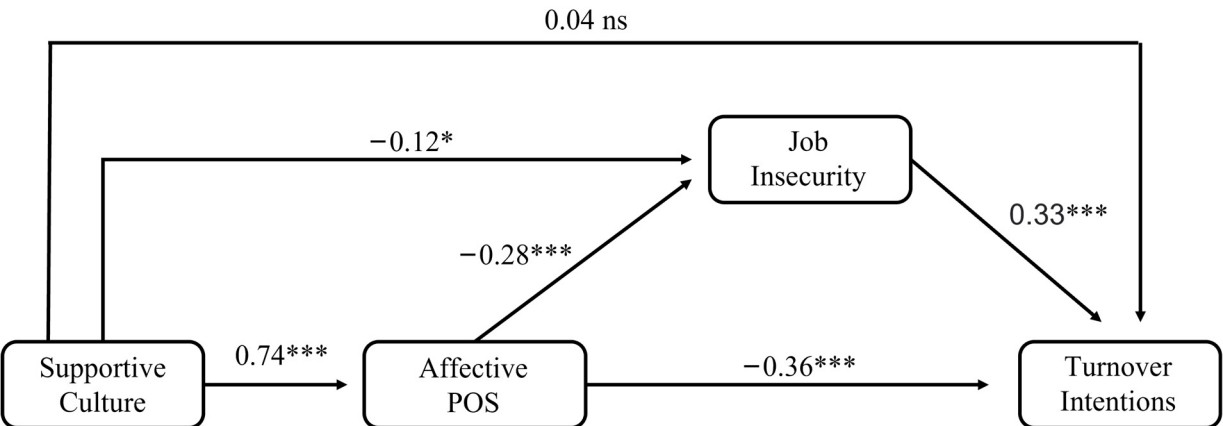

**Figure 3.** Model 1. Notes: ns = No significant; * *p* < 0.05; *** *p* < 0.001.

A significant total indirect effect was also observed since the confidence interval did not contain zero. Again, this indirect effect is divided into three significant indirect effects: the serial indirect effect; the indirect effect in which cognitive POS mediates the relationship between supportive culture and TI; the indirect effect in which JI mediates the relationship between supportive culture and TI (Table 9). When the contrasts were analyzed, it was found that the strongest indirect effect is the one in which JI mediates the relationship between supportive culture and TI. When mediators were introduced in the regression equation, the direct effect of supportive culture on TI continued to be significant, but its intensity decreased, which leads to the conclusion that this is a partial mediation effect (Figure 4).

**Table 9.** Indirect effects of Model 2.

|  | Indirect Effects | |
| --- | --- | --- |
|  | **Estimates** | **95% CI with Bootstrap Correction** |
| Model 2 | | |
| Total | −0.21 (0.27) | [−0.26; −0.15] |
| SC → CPOS → TI | −0.09 (0.02) | [−0.14; −0.05] |
| SC → JI → TI | −0.68 (0.02) | [−0.11; −0.03] |
| SC → CPOS → JI → TI | −0.05 (0.01) | [−0.07; −0.03] |

Note: Total Effect SC → TI = −0.36 (0.04). The standard error is in parentheses. SC = supportive culture; TI = turnover intentions; CPOS = cognitive perceived organizational support; JI = job insecurity.

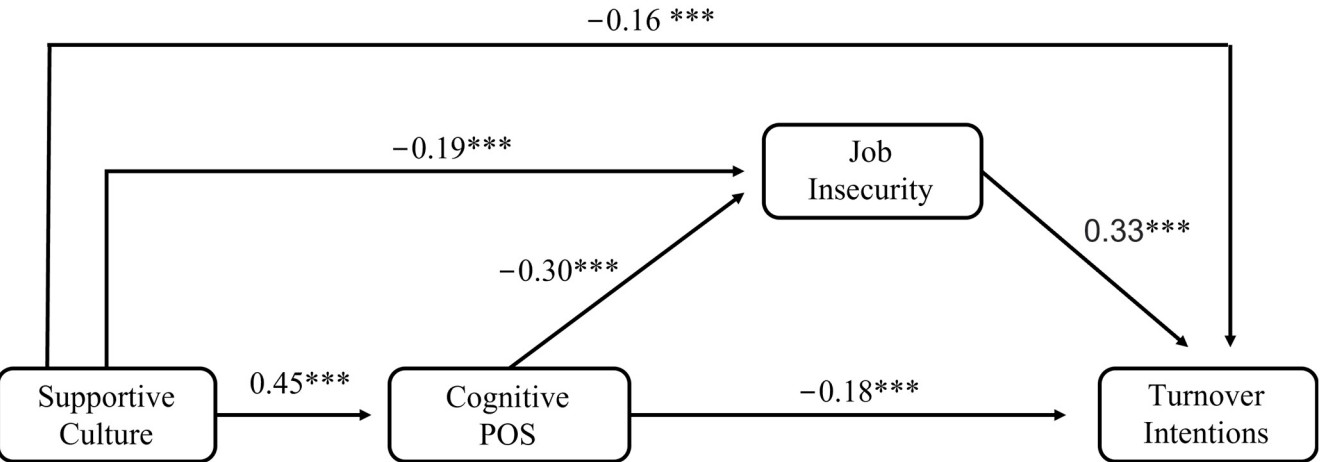

**Figure 4.** Model 2. Note: *** *p* < 0.001.

When testing Hypothesis 6b, a significant total indirect effect was observed since the confidence interval did not contain zero. This indirect effect is divided into three indirect effects, but only two of the effects are significant: the serial indirect effect; the indirect effect in which affective POS mediates the relationship between goal culture and TI (Table 10). When the contrasts were analyzed, it was found that the strongest indirect effect is the one in which affective POS mediates the relationship between goal culture and TI. When mediators were introduced in the regression equation, the direct effect of goal culture on TI was no longer significant, which leads to the conclusion that we are dealing with a total mediation effect (Figure 5).

**Table 10.** Indirect effects of Model 3.

| | Indirect Effects | |
| --- | --- | --- |
| | **Estimates** | **95% CI with Bootstrap Correction** |
| Model 3 | | |
| Total | −0.36 (0.05) | [−0.45; −0.28] |
| GC → APOS → TI | −0.26 (0.04) | [−0.35; −0.18] |
| GC → JI → TI | −0.01 (0.02) | [−0.05; 0.03] |
| GC → APOS → JI → TI | −0.09 (0.02) | [−0.13; −0.06] |

Note: Total Effect GC → TI = −0.36 (0.04). The standard error is in parentheses. GC = goal culture; TI = turnover intentions; APOS = affective perceived organizational support; JI = job insecurity.

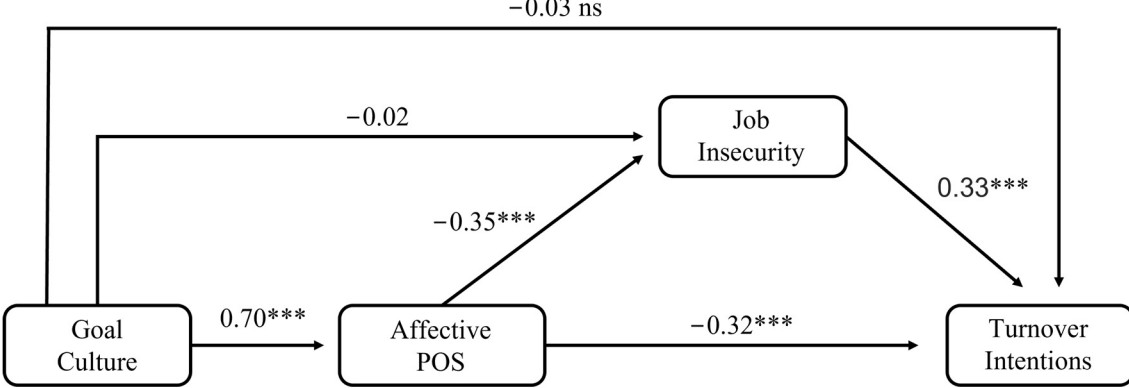

**Figure 5.** Model 3. Note: ns = no significant; *** *p* < 0.001.

A significant total indirect effect was also observed since the confidence interval did not contain zero. Again, this indirect effect is divided into three significant indirect effects: the serial indirect effect; the indirect effect in which cognitive POS mediates the relationship between goal culture and TI; the indirect effect in which JI mediates the relationship between goal culture and TI (Table 11). Then the contrasts were analyzed, and it was found that the strongest indirect effect is the one in cognitive POS mediates the relationship between goal culture and TI. When the mediators were introduced into the regression equation, the direct effect of goal culture on TI continued to be significant, but its intensity decreased, which leads to the conclusion that we are dealing with a partial mediation effect (Figure 6).

**Table 11.** Indirect Effects of Model 4.

| | Indirect Effects | |
|---|---|---|
| | Estimates | 95% CI with Bootstrap Correction |
| Model 4 | | |
| Total | −0.19 (0.03) | [−0.25; −0.14] |
| GC → CPOS → TI | −0.08 (0.02) | [−0.13; −0.04] |
| GC → JI→ TI | −0.05 (0.02) | [−0.09; −0.17] |
| GC → CPOS → JI → TI | −0.05 (0.01) | [−0.08; −0.03] |

Note: Total effect GC → TI = −0.36 (0.04). The standard error is in parentheses. GC = goal culture; TI = turnover intentions; CPOS = cognitive perceived organizational support; JI = job insecurity.

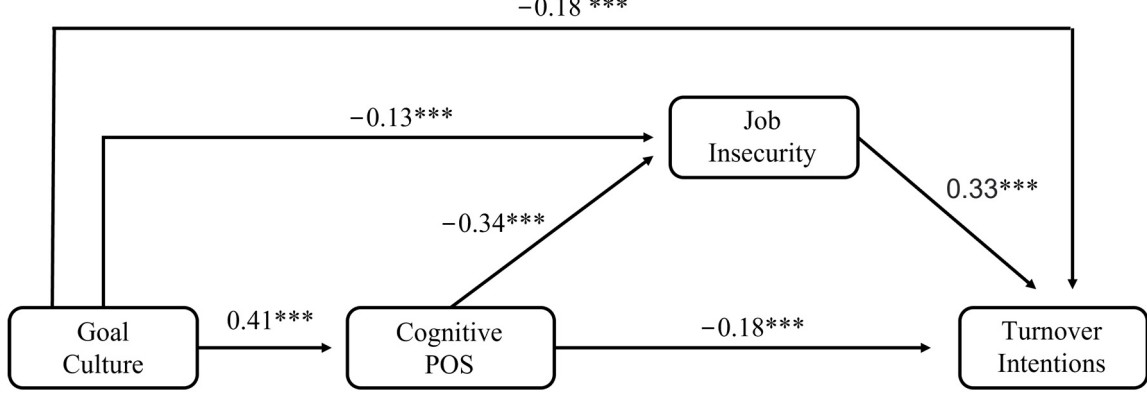

**Figure 6.** Model 4. Note: *** $p < 0.001$.

## 4. Discussion

This study aimed to study the relationship between POC and TI and whether POS and JI are the mechanisms that explain this relationship.

The participants in this study revealed that they have a high POC and POS, as well as low JI and low TI. Among the various types of organizational culture, participants have a lower perception of the culture of innovation. These results align with what Allen et al. (2015) say, that one of the limitations of telework for organizations is the culture of innovation. Participants from the private sector revealed a significantly higher POC than participants from the public sector. It is in line with the results obtained by Rus and Rusu (2015) that employees in the private organization studied had a more positive organizational culture characterized by homogeneity. It was found that employees in the public sector showed fewer TI than employees in the private sector. These results are in line with what the literature tells us, that employees in the private sector have more TI than employees in the public sector (Wang et al. 2012).

First, only supportive and goal cultures were found to have a negative and significant relationship with TI. These results are partially in line with the literature, that POC works as a reducer of IS (Islam et al. 2012). It should be noted that the culture of support is the one with the strongest relationship with TI, which can be interpreted in light of the theory of conservation of resources because the employee perceiving a high supportive culture does not want to lose this resource, and their TI decreases (Hobfoll 1989).

Secondly, it was proved that there is a positive and significant relationship between some POC types and POS. Rule culture has a negative and significant relationship with both affective and cognitive POS. These results also meet what the literature tells us since, according to Kim and Jang (2018), POC has a positive effect on POS. These results can be interpreted in light of the social identity theory, developed by Tajfel (1978), since when an employee perceives himself as a member of the organization to which he belongs, he absorbs its characteristics, including POS. This hypothesis was only partially confirmed

since the culture of innovation is not significantly related to cognitive POS, which is possibly due to the fact that this type of culture was the least perceived by employees.

Thirdly and as expected, the effect of POS (affective and cognitive) on TI was proven, with affective POS having the strongest relationship with TI. These results are also in line with the literature since POS is one of the antecedents of TI (Hui et al. 2007). In a study conducted in China, Jing and Yan (2022) also obtained the same results, that POS reduces TI. These results can be explained based on the premise of social exchanges (Blau 1964) and the norm of reciprocity (Gouldner 1960) because employees, when they perceive that their organization cares about their well-being, certainly want to stay in it. Another theory that may explain these results is the social comparison theory (Adams 1965), according to which employees tend to compare the organization where they work with other organizations and if they perceive that concern for their well-being is higher in the one where they work, their intentions to leave will certainly decrease.

Fourth, and as expected, POS (affective and cognitive) has a negative and significant relationship with JI. This relationship can be interpreted in the light of the human capital theory (Schultz 1961) because when the organization provides the necessary support to employees, it develops their skills, which gives them a lower perception of JI. These results are in line with what the literature tells us: when employees perceive that their organization cares about the support it provides, their JI decreases (Bohle et al. 2018).

Fifth, and as expected, there was a positive and significant relationship between JI and TI. This relationship can be interpreted based on the premise of social exchanges (Blau 1964) because if your organization does not provide you with a perceived sense of job security, as an exchange, your TI increase. These results also confirm what the literature tells us: employees with high levels of JI seek new workplaces more often (Greenhalgh and Rosenblatt 1984). Additionally, according to Obeng et al. (2020), JI potentiates TI, i.e., when employees perceive high job insecurity, their TI increases.

Finally, it was found that POS (affective and cognitive) and JI are the mechanisms that explain the relationship between POC (supportive and goal-oriented) and JI. When an employee perceives a high supportive and goal-oriented culture, their POS (affective and cognitive) become higher (Kim and Jang 2018), decreasing their JI (Bohle et al. 2018) and ultimately decreasing their TI (Obeng et al. 2020), i.e., this association leads to the employee wanting to stay in the organization. The serial mediating effect of POS and JI on the relationship between POC (innovation and rules) and TI was not tested, as these two types of culture were not shown to have a significant association with TI. As mentioned earlier, innovation POC was the least perceived by the participants of this study which may have influenced these results. In turn, the POC of rules had the highest perception, which may also explain these results.

### 4.1. Limitations

This study has some limitations. First, it is a cross-sectional study, and it is impossible to establish causal relationships. The fact that self-report questionnaires were used is another limitation of this study. To reduce the impact of the variance of the common method, we followed several methodological recommendations by Podsakoff et al. (2003).

The moderating effect of the region where the participant resides on the studied relations was not tested as a limitation. Another suggestion would be to study the serial mediation effect, replacing the perception of IL with the perception of employability, based on the previous study by Moreira et al. (2022). It should also be noted that, according to De Cuyper et al. (2008), it will be much more important for the worker to feel that his or her perception of employability is on the rise rather than job security.

### 4.2. Practical Implications

The strength of this study is that it tells us that POS (affective and cognitive) and JI are the mechanisms that explain the relationship between POC (of rules and goals) and TI.

At a time when the way of working has changed entirely due to the COVID-19 pandemic, changing from primarily face-to-face work to online or hybrid work, organizations, in order to retain their best employees, should be concerned with fostering a perception of a higher culture of innovation in them because these are resources that are challenging to imitate and, according to the "Resource-Based View" theory (Afiouni 2007; Barney 1991), are their competitive advantage in the current labor market. As mentioned earlier, one of the limitations for organizations caused by remote work is precisely related to the culture of innovation (Allen et al. 2015).

## 5. Conclusions

It was observed that this study achieved almost all the proposed objectives and that the conclusions obtained contribute to the advancement of the literature and organizations, especially their HRM. This study indicates that organizations should be concerned with the type of organizational culture that prevails. It would be desirable that they foster the existence of a culture of innovation since it was perceived that this type of culture is the least perceived by employees. It was also perceived that the supportive culture is the one that most enhances POS (affective and cognitive) and that it reduces JI and TI. It was concluded that POS (affective and cognitive) and JI are the mechanisms that explain the relationship between POC (supportive and goals) and TI.

Organizations should be concerned with promoting a supportive and goal-oriented culture so that the POS (affective and cognitive) of their employee's increases (Kim and Jang 2018), decreasing their JI (Bohle et al. 2018) and IT (Obeng et al. 2020). Only then will the organization be able to retain its best employees, as this retention is one of the biggest problems facing them (Reiche 2008).

**Author Contributions:** Conceptualization, M.S. and A.M.; methodology, M.S. and A.M.; software, M.S. and A.M.; validation, M.S., A.M., and L.P.; formal analysis, A.M..; investigation, M.S. and L.P..; resources, M.S.; data curation, A.M.; writing—original draft preparation, M.S. and L.P.; writing—review and editing, A.M.; visualization, M.S.; supervision, A.M.; project administration, A.M.; funding acquisition, A.M. All authors have read and agreed to the published version of the manuscript.

**Funding:** This research received no external funding.

**Institutional Review Board Statement:** Ethical review and approval were waived for this study because all participants, before answering the questionnaire, had to read the informed consent and agree to it. It was the only way they could answer the questionnaire. Participants were informed about the purpose of the study and that the results were confidential, as individual results would never be known but would only be analyzed in the set of all participants.

**Informed Consent Statement:** Written informed consent was obtained from the patient(s) to publish this paper.

**Data Availability Statement:** The data presented in this study are available on request from the corresponding authors. The data are not publicly available since, in their informed consent, participants were informed that the data were confidential and that individual responses would never be known, as data analysis would be of all participants combined.

**Conflicts of Interest:** The authors declare no conflict of interest.

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
