# Peer review of "Perceived Organizational Culture and Turnover Intentions: The Serial Mediating Effect of Perceived Organizational Support and Job Insecurity"

_socsci, doi:10.3390/socsci11080363_

Round 1

Reviewer 1 Report

Thank you for the opportunity to review the manuscript entitled: "Perceived organizational culture and turnover intentions: The 2 serial mediating effect of perceived organizational support and 3 job insecurity".

I would like to congratulate the authors for writing this article. This manuscript is interesting, well written, organized and the choices made in the research seem to be relevant.

However, for publications, it is recommended to explore whether there is more recent research on the topic. I hope these suggestion can be useful to improve your study. Good luck!

Author Response

Dear Reviewer,

Firstly, we would like to thank you for taking the time and effort necessary to provide insightful guidance, which has contributed to improving this new version of the paper. We carefully considered the comments provided. Herein, we explain how we revised the manuscript based on those comments and recommendations.

We appreciate your preliminary comments that will complement our work.

Comment 1: However, for publications, it is recommended to explore whether there is more recent research on the topic. I hope these suggestion can be useful to improve your study. Good luck!

Thank you for the suggestion. Some more relevant references were added as you suggested.

In closing, we would like to thank you again for the comments. We hope that we have dealt with your suggestions satisfactorily and made all the adjustments requested, both in form and substance.

Yours sincerely,

On behalf of my co-authors,

References added to the manuscript:

Çakmak-OtluoÄŸlu, K. Ö., & Ünsal-Akbiyik, B. S. 2015. Perceived job insecurity, affective and normative commitment: The moderating effect of organizational career development opportunities. Psihologia Resurselor Umane Revista AsociaÅ£iei de Psihologie Indusstrială ÅŸi OrganizaÅ£ională, 13(2), 179–188.

De Witte, H., Vander Elst, T., & De Cuyper, N. 2015. Job Insecurity, Health and Well-Being. Sustainable Working Lives, 109–128. doi:10.1007/978-94-017-9798-6_7 

Jing, J.; Yan, J. 2022. Study on the Effect of Employees’ Perceived Organizational Support, Psychological Ownership, and Turnover Intention: A Case of China’s Employee. Int. J. Environ. Res. Public Health, 19, 6016. https:// doi.org/10.3390/ijerph19106016

Ma, B., Liu, S., Lassleben, H. and Ma, G. 2019. The relationships between job insecurity, psychological contract breach and counterproductive workplace behavior: Does employment status matter?. Personnel Review, 48 (2), 595-610. https://doi.org/10.1108/PR-04-2018-0138

Mihajlov, S. and Mihajlov, N. (2016). Comparing public and private employees' job satisfaction and turnover intention. MEST Journal, 4 (1), 75-86. Doi: 10.12709/mest.04.04.01.08

Piccoli, B., Reisel, W. D. and De Witte, H. 2019. Understanding the Relationship Between Job Insecurity and Performance: Hindrance or Challenge Effect?. Journal of Career Development, (2) 089484531983318–. doi:10.1177/0894845319833189 

Rus, M. and Rusu, D. O. 2015. The Organizational Culture in Public and Private Institutions. Procedia - Social and Behavioral Sciences, 187(), 565–569. doi:10.1016/j.sbspro.2015.03.105 

Wang, Y.-D.; Yang, C.; Wang, K.-Y. 2012. Comparing Public and Private Employees' Job Satisfaction and Turnover. Public Personnel Management, 41(3), 557–573. doi:10.1177/009102601204100310 

Reviewer 2 Report

The topic is very interesting and relevant. Below some feed-back to further improve the quality of the paper.

"1.1. Perceived Organisational Culture" authors refer to two main theories; it would be better clarify  why the authors mentioned these two specific theories and whether and how they integrate them with each other for the research work

"1.2. Turnover Intentions" it would be interesting if the authors would integrate the info on the topic of turnover with respect to different organizational contexts and (if possible) find data about the turnover trends in the two main field of the study (public and private sector)

"1.4 Job Insecurity" should be integrated with further references

"Instruments" - Please explain the reasons for the choice of scales proposed in the survey

"Results" - I think it'd be interesting explore if there are any differences between public and private sector

"Discussion" - I think it'd be important expand the reasoning about the result

Author Response

Dear Reviewer,

Firstly, we would like to thank you for taking the time and effort necessary to provide insightful guidance, which has contributed to improving this new version of the paper. We carefully considered the comments provided. Herein, we explain how we revised the manuscript based on those comments and recommendations.

We appreciate your preliminary comments that will complement our work.

We are very thankfull for all interesting insights.

Comment 1: Perceived Organisational Culture" authors refer to two main theories; it would be better clarify why the authors mentioned these two specific theories and whether and how they integrate them with each other for the research work

Thank you for your comment. Schein's Theory was addressed as it has relevance in interpreting the relationship of culture with the perception of organisational support.

The Cameron and Quinn theory was addressed and explained because the questionnaire applied to assess culture is based on this Theory.

In the introduction the following sentences were added:

In this study, we will address two theories of culture: Schein's Theory and Cameron and Quinn's Theory. Schein's Theory will be addressed due to its relevance in interpreting the relationship between culture and the perception of organisational support. Cameron and Quinn's Theory will be addressed and explained because the questionnaire applied to assess culture is based on this Theory.

Comment 2: Turnover Intentions" it would be interesting if the authors would integrate the info on the topic of turnover with respect to different organizational contexts and (if possible) find data about the turnover trends in the two main field of the study (public and private sector).

We appreciate your contributions and added two studies from different countries on the differences between the public and private sectors.

Comment 3: Job Insecurity" should be integrated with further references.

We thank you for your suggestions. We have added new references in job insecurity.

Comment 4: "Instruments" - Please explain the reasons for the choice of scales proposed in the survey.

We thank you for your suggestions. At the beginning of the sub-chapter on instruments, we added the reason for their choice, which was the fact that they are instruments already adapted to the Portuguese population.

Comment 5: Results" - I think it'd be interesting explore if there are any differences between public and private sector

We thank you for your suggestions. At the beginning of the sub-chapter on instruments, we added the reason for their choice, which was the fact that they are instruments already adapted to the Portuguese population.

Comment 6: "Discussion" - I think it'd be important expand the reasoning about the result.

We thank reviewer 2 for his/her suggestions. We added in the discussion some explanations for the results.

In closing, we would like to thank you again for the comments. We hope that we have dealt with your suggestions satisfactorily and made all the adjustments requested, both in form and substance.

Yours sincerely,

On behalf of my co-authors,

References added to the manuscript:

Çakmak-OtluoÄŸlu, K. Ö., & Ünsal-Akbiyik, B. S. 2015. Perceived job insecurity, affective and normative commitment: The moderating effect of organizational career development opportunities. Psihologia Resurselor Umane Revista AsociaÅ£iei de Psihologie Indusstrială ÅŸi OrganizaÅ£ională, 13(2), 179–188.

De Witte, H., Vander Elst, T., & De Cuyper, N. 2015. Job Insecurity, Health and Well-Being. Sustainable Working Lives, 109–128. doi:10.1007/978-94-017-9798-6_7 

Jing, J.; Yan, J. 2022. Study on the Effect of Employees’ Perceived Organizational Support, Psychological Ownership, and Turnover Intention: A Case of China’s Employee. Int. J. Environ. Res. Public Health, 19, 6016. https:// doi.org/10.3390/ijerph19106016

Ma, B., Liu, S., Lassleben, H. and Ma, G. 2019. The relationships between job insecurity, psychological contract breach and counterproductive workplace behavior: Does employment status matter?. Personnel Review, 48 (2), 595-610. https://doi.org/10.1108/PR-04-2018-0138

Mihajlov, S. and Mihajlov, N. (2016). Comparing public and private employees' job satisfaction and turnover intention. MEST Journal, 4 (1), 75-86. Doi: 10.12709/mest.04.04.01.08

Piccoli, B., Reisel, W. D. and De Witte, H. 2019. Understanding the Relationship Between Job Insecurity and Performance: Hindrance or Challenge Effect?. Journal of Career Development, (2) 089484531983318–. doi:10.1177/0894845319833189 

Rus, M. and Rusu, D. O. 2015. The Organizational Culture in Public and Private Institutions. Procedia - Social and Behavioral Sciences, 187(), 565–569. doi:10.1016/j.sbspro.2015.03.105 

Wang, Y.-D.; Yang, C.; Wang, K.-Y. 2012. Comparing Public and Private Employees' Job Satisfaction and Turnover. Public Personnel Management, 41(3), 557–573. doi:10.1177/009102601204100310 

Reviewer 3 Report

the study was well-done and useful for future research studies.

writing issues:

page 1 line 5 'This study aims' -- the abstract should state what was done rather than the aim, which might be appropriate for a proposal

page 1 lines 18-19 seen redundant to the next paragraph. The manuscript would benefit from a detailed copy review/edit

Author Response

Dear Reviewer,

Firstly, we would like to thank you for taking the time and effort necessary to provide insightful guidance, which has contributed to improving this new version of the paper. We carefully considered the comments provided. Herein, we explain how we revised the manuscript based on those comments and recommendations.

We appreciate your preliminary comments that will complement our work.

Comment 1: page 1 line 5 'This study aims' -- the abstract should state what was done rather than the aim, which might be appropriate for a proposal

We thank you for your comments. We have added in summary the hypotheses formulated in this study.

Comment 2: page 1 lines 18-19 seen redundant to the next paragraph. The manuscript would benefit from a detailed copy review/editç

We thank you for your suggestions. These two sentences "affective POS and JI have a serial mediation effect in the relationship between supportive and goal POC and TI; cognitive POS and JI have a serial mediation effect in the relationship between goal POC and TI" seem redundant, but they are not and so we have kept both sentences.

In closing, we would like to thank you again for the comments. We hope that we have dealt with your suggestions satisfactorily and made all the adjustments requested, both in form and substance.

Yours sincerely,

On behalf of my co-authors,

References added to the manuscript:

Çakmak-OtluoÄŸlu, K. Ö., & Ünsal-Akbiyik, B. S. 2015. Perceived job insecurity, affective and normative commitment: The moderating effect of organizational career development opportunities. Psihologia Resurselor Umane Revista AsociaÅ£iei de Psihologie Indusstrială ÅŸi OrganizaÅ£ională, 13(2), 179–188.

De Witte, H., Vander Elst, T., & De Cuyper, N. 2015. Job Insecurity, Health and Well-Being. Sustainable Working Lives, 109–128. doi:10.1007/978-94-017-9798-6_7 

Jing, J.; Yan, J. 2022. Study on the Effect of Employees’ Perceived Organizational Support, Psychological Ownership, and Turnover Intention: A Case of China’s Employee. Int. J. Environ. Res. Public Health, 19, 6016. https:// doi.org/10.3390/ijerph19106016

Ma, B., Liu, S., Lassleben, H. and Ma, G. 2019. The relationships between job insecurity, psychological contract breach and counterproductive workplace behavior: Does employment status matter?. Personnel Review, 48 (2), 595-610. https://doi.org/10.1108/PR-04-2018-0138

Mihajlov, S. and Mihajlov, N. (2016). Comparing public and private employees' job satisfaction and turnover intention. MEST Journal, 4 (1), 75-86. Doi: 10.12709/mest.04.04.01.08

Piccoli, B., Reisel, W. D. and De Witte, H. 2019. Understanding the Relationship Between Job Insecurity and Performance: Hindrance or Challenge Effect?. Journal of Career Development, (2) 089484531983318–. doi:10.1177/0894845319833189 

Rus, M. and Rusu, D. O. 2015. The Organizational Culture in Public and Private Institutions. Procedia - Social and Behavioral Sciences, 187(), 565–569. doi:10.1016/j.sbspro.2015.03.105 

Wang, Y.-D.; Yang, C.; Wang, K.-Y. 2012. Comparing Public and Private Employees' Job Satisfaction and Turnover. Public Personnel Management, 41(3), 557–573. doi:10.1177/009102601204100310